

# Transcriptome and metabolome response of eggplant against *Ralstonia solanacearum* infection

Xi Ou Xiao[1,2,3], Wenqiu Lin[1,2,3], Enyou Feng[4] and Xiongchang Ou[1,2,3]

[1] South Subtropical Crop Research Institute Chinese Academy of Tropical Agricultural Sciences (CATAS), Zhanjiang, Guangdong, China
[2] Key Laboratory of Hainan Province for Postharvest Physiology and Technology of Tropical Horticultural Products, Zhanjiang, Guagndong, China
[3] Zhanjiang City Key Laboratory for Tropical Crops Genetic Improvement, Zhanjiang, Guangdong, China
[4] Zhanjiang Academy of Agricultural Sciences, Zhanjiang, Guangdong, China

## ABSTRACT

Bacterial wilt is a soil-borne disease that represents ubiquitous threat to *Solanaceae* crops. The whole-root transcriptomes and metabolomes of bacterial wilt-resistant eggplant were studied to understand the response of eggplant to bacterial wilt. A total of 2,896 differentially expressed genes and 63 differences in metabolites were identified after inoculation with *Ralstonia solanacearum*. Further analysis showed that the biosynthesis pathways for phytohormones, phenylpropanoids, and flavonoids were altered in eggplant after inoculation with *R. solanacearum*. The results of metabolomes also showed that phytohormones played a key role in eggplant response to bacterial wilt. Integrated analyses of the transcriptomic and metabolic datasets indicated that jasmonic acid (JA) content and gene involved in the JA signaling pathway increased in response to bacterial wilt. These findings remarkably improve our understanding of the mechanisms of induced defense response in eggplant and will provide insights intothe development of disease-resistant varieties of eggplant.

## INTRODUCTION

Eggplant (*Solanum melongena* L.) is an important vegetable in tropical and subtropical areas. According to the Food and Agriculture Organization (FAO), 55,197,878 tons of eggplant fruit were produced worldwide in 2019. Eggplant fruit contains a variety of nutrients, such as vitamins, phenolics, and antioxidants, which are beneficial to human health (*Gurbuz et al., 2018*). At the same time, diseases, such as bacterial wilt (*Barik et al., 2020*) and verticillium wilt (*Yang et al., 2019*), can lead to significant yield loss.

Bacterial wilt is a soil-borne disease caused by the pathogen *R. solanacearum*. This pathogen has more than 450 host plant species, which belong to 54 families (*Jiang et al., 2017*). Bacterial wilt is one of the most destructive plant diseases because it is difficult to control and can cause considerable production losses (*Jiang et al., 2017*; *Barik et al., 2020*). To date, no effective chemical management strategy is available for bacterial wilt disease. In field practice, the management and control of bacterial wilt include resistant cultivar

Corresponding author
Xi Ou Xiao,
xiao-forlearning@catas.cn

selection (*Barik et al., 2020*), grafting (*Manickam et al., 2021*), crop rotation (*Ayana & Fininsa, 2017*), and antagonistic organisms (*Yuliar, Nion & Toyota, 2015*). Among these management practices, the resistant cultivar selection is the most economical and efficient means.

Several QTLs (quantitative trait locus) relate to bacterial wilt resistance have been identified in different plants, such as eggplant (*Salgon et al., 2017*, *2018*), tomato (*Kim et al., 2018*; *Abebe et al., 2020*), potato (*Habe et al., 2019*), and peanut (*Wang et al., 2018*; *Luo et al., 2019*). These QTLs provide a good foundation for molecular marker-assisted selection and gene editing for breeding resistant cultivars. However, the mechanisms which plants regulate defense responses remain limited. How plants respond to bacterial wilt should be understood to breed resistant varieties efficiently and improve their coping ability.

Once a plant detects pathogen invasion, the plant initiates a defense response against the disease, including the expression of defense genes and biosynthesis of secondary metabolites. Secondary metabolites, such as alkaloids, flavonoids, and phenolics, have been reported to play a key role in plant defense reactions (*Zaynab et al., 2018*). Additionally, phytohormones, such as salicylic acid (SA), JA, and ethylene (ET), and their signaling pathways play a key role in the plant disease defense response (*Dong, 1998*).

To understand the mechanism of bacterial wilt resistance, RNA-seq was used to characterize the transcriptome changes after inoculation with *R. solanacearum* (*Chen, Ma & Chen, 2019*; *Ishihara et al., 2012*; *Jiang et al., 2019*; *Li et al., 2021c*; *Zhao et al., 2019*; *Zuluaga et al., 2015*). The results showed that a set of genes is remarkable differentially expressed after the plant was attacked by *R. solanacearum*. For example, 9,831 DEGs, including *WRKY* transcription factors, *ERFs* transcription factors, and defense-related genes, in tobacco responded to *R. solanacearum* infection. The Kyoto Encyclopedia of Genes and Genomes (KEGG) analysis demonstrated phenylpropane pathways as primary resistance pathways to *R. solanacearum* infection (*Li et al., 2021c*). In Arabidopsis roots, 2,698 DEGs were identified after *R. solanacearum* infection. The DEGs involved in multiple-hormone signaling cascades include abscisic acid (ABA), auxin, JA and ET. In *Casuarina equisetifolia*–*R. solanacearum* interaction, 479 DEGs, which are classified into brassinosteroid, SA, and JA signaling pathways, are detected (*Wei et al., 2021*).

Except transcriptomics, other omics, such as proteomics and metabolomics, have been widely used to analyze plant biotic and abiotic stress responses. Metabolomics is focuses on all small molecular components and is widely used to study plant biological function and mechanism. Metabolomics is attaining increasing attention in pathogen–plant interaction to elucidate plant defense mechanisms (*Shulaev et al., 2008*; *Chen, Ma & Chen, 2019*; *Schauer & Fernie, 2006*). Multiomics data especially combined metabolomic and transcriptomic analysis, have been integrated and analyzed to understand the complex signaling pathways involved in plant defense reactions (*Yuan et al., 2018*; *Sun et al., 2020*; *Li et al., 2021b*; *Wei et al., 2021*).

Genetic resources for bacterial wilt resistance in eggplant are available and the inheritance of bacterial wilt resistance is complex and which based on the *R. solanacearum* race and environment (*Barik et al., 2020*). RNA-seq results showed that eggplant resistance
to bacterial wilt involved in cell death pathways, cell receptor signaling pathways (*Chen et al., 2018*). *Peng et al. (2021)* showed that the *MAPK* signaling pathway, plant pathogen interaction, and glutathione metabolism were co-enriched in roots and stems after inoculation with *R. solanacearum*. Further research showed that SmRPP13L4 positively regulated the resistance of eggplant to bacterial wilt (*Peng et al., 2021*). Other researches showed that the *NAC* (*Chen et al., 2016*), *MYB44* (*Qiu et al., 2019*) and *TCP7a* (*Xiao et al., 2022*) transcriptional factor also regulated the bacterial wilt resistance.

In this study, we performed comparative transcriptomic and metabolomic analyses after *R. solanacearum* inoculation into bacterial wilt-resistant eggplant to understand the defense responses of eggplant against bacterial wilt. *R. solanacearum*-induced DEGs and metabolites were identified. These results extend our understanding of the molecular mechanism of eggplant response to *R. solanacearum*.

## MATERIALS AND METHODS

### Plant material

Eggplant inbred lines "NY-1" (R genotype, highly resistant to bacterial wilt) and "KY" (S genotype, highly susceptible to bacterial wilt) were obtained from the South Subtropical Crop Research Institute Chinese Academy of Tropical Agricultural Sciences. Seeds were sown in 15 cm diameter pots. The growing material was placed in pots and composed of sterile vermiculite and clay mixed in a 3:1 volume/volume ratio. Seedlings were grown under 28 °C/25 °C day/night with a 16 h light/8 h dark photoperiod condition. After 4 weeks of culture, when seedlings were at the 4-leaf stage, the culture was incubated with *R. solanacearum*.

### Bacterial strain and inoculation

The *R. solanacearum* strain GMI1000-tac-EGFP was grown overnight on 2, 3, 5-triphenyl tetrazolium chloride medium at 28 °C and suspended in sterile distilled water (*Xiao et al., 2021*). The suspension was adjusted to 0.12 ($10^8$ colony-forming units/ml) at 600 nm. The roots of eggplants were cut at 0–1 cm from the apex and then inoculated in 50 ml suspended *R. solanacearum*. After inoculation with *R. solanacearum*, plants were grown under 30 °C/32 °C day/night with a 16 h light/8 h dark photoperiod condition. Disease was rated on a scale of 0 to 4: 0 = no symptoms, 1 = 0–25% leaves wilted, 2 = 25–50% leaves wilted, 3 = 50–75% leaves wilted, 4 = 75–100% wilted and plant dead. The disease index (DI) was calculated using the formula: DI = ((N0 × 0 + N1 × 1 + N2 × 2 + N3 × 3 + N4 × 4)/(total number of plants)). N0 to N4 were the number of plants with disease rating scale values of 0 to 4, respectively. The EGFP fluorescence of *R. solanacearum* was detected using LUYOR-3415.

### RNA-seq

At 0, 24, and 48 hpi, the roots of 10 eggplants were collected, mixed, immediately frozen in liquid nitrogen, and stored at −80 °C. Three biological replicates were established for each treatment group. Total RNA was extracted using the Spin Column Plant total RNA Purification Kit (Sangon Biotech, Shanghai, China) following the manufacturer's protocol.

RNA quantification was performed using the Qubit RNA Assay Kit in Qubit 2.0 Fluorometer. RNA integrity was assessed using the RNA Nano 6000 Assay Kit of the Agilent Bioanalyzer 2100 system. After the Illumina sequencing libraries were established, cDNA libraries were sequenced on the Illumina HiSeq platform. The mRNA-Seq was assembled and analyzed by the Guangzhou Gene Denovo Biotechnology Corporation (Guangzhou, China). The statistical power of this experimental design is 0.9; values for alpha and CV were 0.05, which were calculated in RNASeqPower. The effect parameter was 2. The sample size results at 6× sequencing depths were 7.55.

The low-quality reads were filtered using fastp (v0.19.3) with the following standard: the N content in any sequencing reads exceeded 10% of the base number of the read and the low quality (Q<=20) bases contained in reads exceeded 50%. The cleaned reads were aligned to the eggplant reference genome (eggplant genome consortium V3) (*Barchi et al., 2019*) using HISAT2 (version 2.1.0; http://daehwankimlab.github.io/hisat2/). The reads count of each gene was calculated by featureCounts (v1.6.2; *Liao, Smyth & Shi, 2014*). The differential expression analysis of two groups was performed using the DESeq2 R package (V1.22.1; *Love, Huber & Anders, 2014*). For identifying DEGs, absolute fold change ≥ 2 and false discovery rate (FDR) < 0.01 were used as screening criteria. The expression patterns of DEGs were analyzed using the Mfuzz R package (*Kumar & Futschik, 2007*). The heatmap was analyzed by the pheatmap R package.

## Metabolite profiling using UPLC-MS/MS

The freeze-dried eggplant root was crushed using a mixer mill (MM 400; Retsch, Haan, Germany) with a zirconia bead for 1.5 min at 30 Hz. Approximately 100 mg lyophilized powder was dissolved 1.2 ml of 70% methanol solution. After the solution was vortexed for 30 s every 30 min six times, the solution was placed in a refrigerator at 4 °C overnight. Then the solutions were centrifuged at 12,000 rpm for 10 min and the supernatant fluid was filtered (SCAA-104, 0.22 μm pore size) before UPLC-MS/MS analysis.

## UPLC conditions

Sample extracted supernatant fluid was separated using an UPLC system (UPLC, SHIMADZU Nexera X2). The chromatographic column was Agilent SB-C18 (1.8 μm, 2.1 mm × 100 mm) and the column temperature was at 40 °C. The flow rate was 0.35 ml/min with a total injection volume of 4 μl. The mobile phase A was pure water with 0.1% formic acid and the mobile phase B was acetonitrile with 0.1% formic acid. Within 0–9 min, the ratio of A to B was 95:5 → 5:95 and kept for 1 min. Within 10.00–11.10 min, the ratio of A to B was 5:95 → 95:5 and kept for 3 min. The effluent was alternatively connected to an ESI-triple quadrupole-linear ion trap (QTRAP)-MS.

## ESI-Q TRAP-MS/MS

The effluent was analyzed by an AB4500 Q TRAP UPLC/MS/MS System which equipped with an ESI Turbo Ion-Spray interface. The ESI Turbo Ion-Spray interface was controlled by the Analyst 1.6.3 software (AB Sciex, Framingham, MA, USA). The ESI source operation parameters were as follows: the ion source was turbo spray; positive ions mode

was 5,500 V and the negative ion mode was −4,500 V; the ion source gas I, gas II, and curtain gas were set to 50, 60, and 25.0 psi, respectively. Ten and 100 µmol/l polypropylene were used for instrument tuning and mass calibration for triple quadrupole and LIT modes, respectively. Triple quadrupole analysis was performed under a multiple reaction monitoring model with the collision gas (nitrogen) set to medium. After the declustering potential and collision energy optimization, the declustering potential and collision energy for individual multiple reaction monitoring transitions were obtained. Based on the metabolites eluted within each period, a specific set of multiple reaction monitoring transitions was monitored.

## Data analysis

Quality control (QC) samples were prepared by mixing sample extracts. A QC sample was inserted into each of the 10 detected samples during the stability evaluation of analytical conditions. The metabolites was identified in accordance with the metware database and quantified by multiple reaction monitoring.

Significantly regulated metabolites between groups were determined by VIP ≥ 1 and absolute $Log_2FC ≥ 1$. VIP values were extracted from OPLS-DA results, which also contained score and permutation plots generated using the R package MetaboAnalystR (*Chong & Xia, 2018*). Data were subjected to log transformation (log2) and mean centering before OPLS-DA. A permutation test (200 permutations) was performed to avoid overfitting. The gene-metabolite correlation networks were analyzed by differential genes and differential metabolites with Pearson correlation coefficient > 0.80 and $p$-value < 0.05 in each pathway.

## KEGG annotation and enrichment analyses

KEGG annotation of the identified metabolites was performed using the KEGG compound database (http://www.kegg.jp/kegg/compound/). Then, the annotated metabolites were mapped to the KEGG pathway database. The differential metabolites were classified according to the types of pathways in KEGG according to the annotation of KEGG.

# RESULTS

## Analysis of the bacterial wilt resistance of eggplant

After the R and S genotypes of eggplant material were inoculated with the GMI1000-tac-EGFP strain, the DI and EGFP fluorescence were analyzed. The results showed that at 10 days after inoculation with *R. solanacearum*, the DIs of R and S were 0 and 2.48, respectively (Fig. 1A). R genotypes showed normal phenotype, whereas S genotypes showed wilt (Fig. 1B). EGFP fluorescence was detected in S stem and root but was not observed in the R stem and root (Figs. 1C and 1D). This result showed that R genotypes were highly resistant to GMI1000.

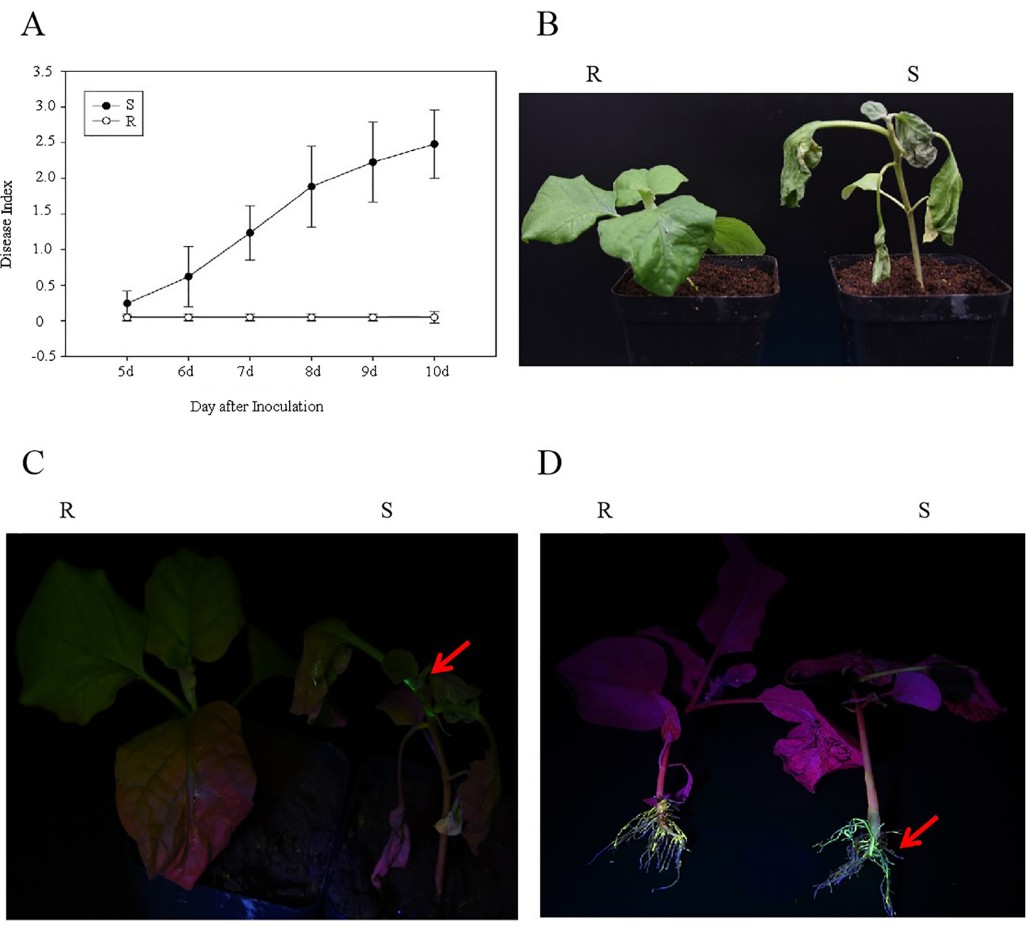

**Figure 1 Analysis of eggplant bacterial wilt resistance.** (A) Disease index of R and S eggplant inoculated with *R. solanacearum*. (B) Phenotype of wilt after inoculation with *R. solanacearum* at 10 days. (C) EGFP fluorescence results at stem. (D) EGFP fluorescence results at the root. The red arrow indicates the EGFP fluorescence.

## Induced responses to bacterial wilt in global transcriptome changes of eggplant

The transcriptome was compared after inoculation of GMI1000-tac-EGFP to understand the mechanism of the bacterial wilt defense response of eggplant. Three time points (*i.e.*, 0, 24, and 48 h postinoculation (hpi)) were analyzed.

Approximately, 390.98 million cleaned reads were generated for nine samples (Table 1). About 85% cleaned reads were aligned to the eggplant reference genome. Transcriptomic sequences were deposited in the NCBI Sequence Read Archive under accession number PRJNA837016.

PCA showed that the first two PCAs explained 59.48% of the total variation (Fig. 2A). The heatmap of DEGs showed a significant difference in gene expression level after the inoculation of *R. solanacearum* (Fig. 2B). After filtration by FDR < 0.01 and absolute Log2 (fold change (FC)) ≥ 1, 1,831 (799 upregulated and 1,032 downregulated), 1,416 (708 upregulated and 708 downregulated), and 1,032 (538 upregulated and 494 downregulated) DEGs were identified in R-0h_vs_R-24h, R-0h_vs_R-48h, and R-24h_vs_R-48h,

**Table 1  Summary of RNA-seq and mapping results.**

| Sample | Clean reads | Reads mapped | Unique mapped |
|---|---|---|---|
| R0h-1 | 44682626 | 37833014 (84.67%) | 35475538 (79.39%) |
| R0h-2 | 49200972 | 41973818 (85.31%) | 39310684 (79.90%) |
| R0h-3 | 43577276 | 37139673 (85.23%) | 34776165 (79.80%) |
| R24h-1 | 42181106 | 35684002 (84.60%) | 33436786 (79.27%) |
| R24h-2 | 47172092 | 39987170 (84.77%) | 37498014 (79.49%) |
| R24h-3 | 41443394 | 35095278 (84.68%) | 32848381 (79.26%) |
| R48h-1 | 39695734 | 33712594 (84.93%) | 31702472 (79.86%) |
| R48h-2 | 42944098 | 36503682 (85.00%) | 34249420 (79.75%) |
| R48h-3 | 40078694 | 34034257 (84.92%) | 31930569 (79.67%) |

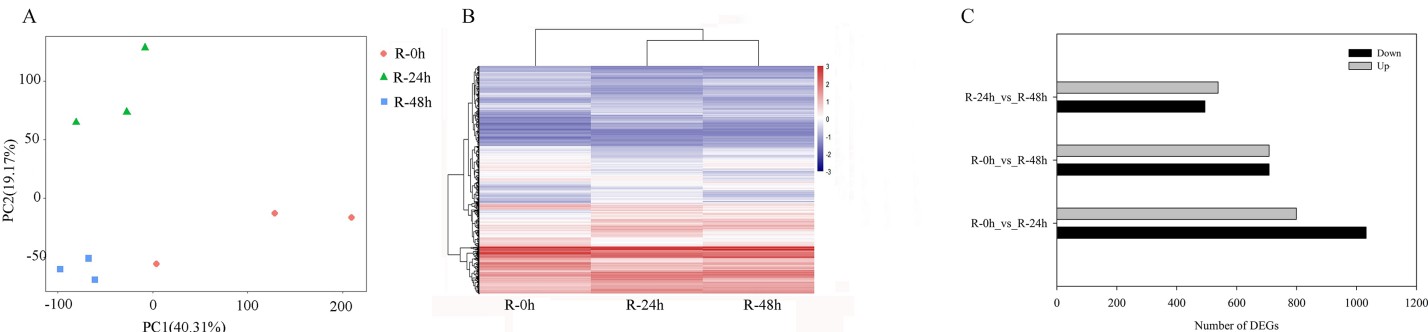

**Figure 2  Differential gene expression of eggplant response to bacterial wilt.** (A) Principal component analysis of all transcripts (RPKM values) detected in root. Data points represent different samples. (B) Clustering analysis and heat map of expression measures of DEGs detected in each of the experimental conditions. (C) Numbers of upregulated and downregulated genes after inoculation with bacterial wilt over time.

respectively (Fig. 2C). These DEGs are listed in Tables S1–S3. After the DEGs were selected as absolute $Log_2 > 5$, 27 DEGs were selected. In the R-0h_Vs_R-24h group, eight genes were unregulated and nine genes were downregulated. In R-0h_Vs_ R-48h, the 11 DEGs were downregulated. But in the R-24h_Vs_ R48-h, there were only one downregulated gene (Table S4).

A total of 485 and 751 genes were upregulated and downregulated, respectively, in at least one time point. Four genes were common in the R-0h_vs_R-24hUp and R-0h_vs_R-48hDn sets, and a gene was common in the R-0h_vs_R-24hDn and R-0h_vs_R-48hUp sets (Fig. 3).

## KEGG and KOG classification of DEGs

DEGs were mapped to the KEGG pathway to understand DEGs function in the eggplant defense response. The top 20 pathways included metabolic pathways, biosynthesis of secondary metabolites, plant hormone signal transduction, *MAPK* signaling pathway, plant–pathogen interaction, and flavonoid biosynthesis pathway (Fig. S1).

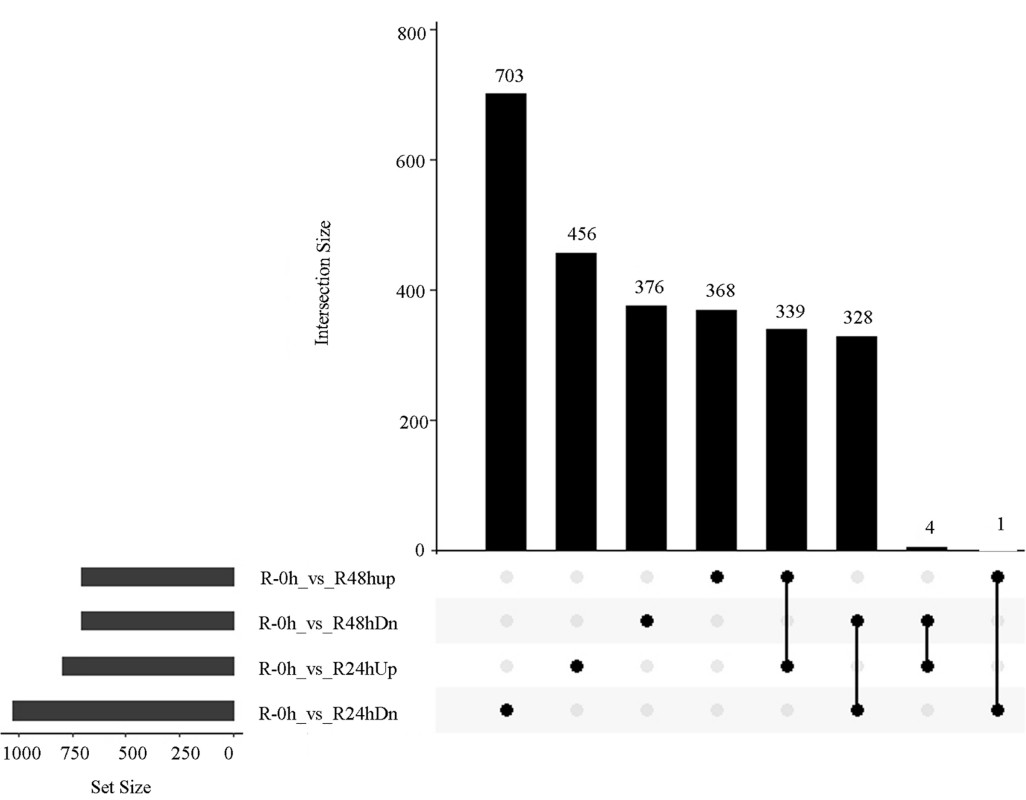

**Figure 3** **UpSet plot showing overlap of upregulated and downregulated genes of eggplant response to bacterial wilt.**

## Expression pattern analysis of DEGs

A set of genes with similar expression patterns was functionally correlated. Six expression patterns were obtained in accordance with the DEGs data. The expression pattern of the centers of cluster 1 was upregulated at 24 and 48 hpi. The expression pattern of the centers of cluster 2 was downregulated at 24 hpi and then upregulated at 48 hpi. Ultimately, the expression level at 0 and 48 hpi were not significantly different. The expression pattern of the centers of cluster 3 was upregulated at 24 hpi and then downregulated at 48 hpi. Ultimately, the expression level at 0 and 48 hpi was not ignificantly different.

The expression pattern of the centers of cluster 4 was upregulated at 24 hpi and then downregulated at 48 hpi. However, the expression level at 48 hpi was significantly increased compared with that at 0 hpi. The expression pattern of the centers of cluster 5 was downregulated at 24 hpi and then upregulated at 48 hpi. However, the expression level at 48 hpi was significantly decreased compared with that at 0 hpi. The expression pattern of the centers of cluster 6 was downregulated at 24 and 48 hpi (Fig. 4 and Table S5). A total of 488 (cluster 1) and 443 (cluster 6) DEGs were maintained to be upregulated and downregulated at 24 and 48 hpi. In cluster 1, there were seven *WRKY* transcriptional factors, four *TIFY* transcriptional factors and three *bHLH* transcriptional factors. Meanwhile, there were seven resistance genes were identified (Table 2).
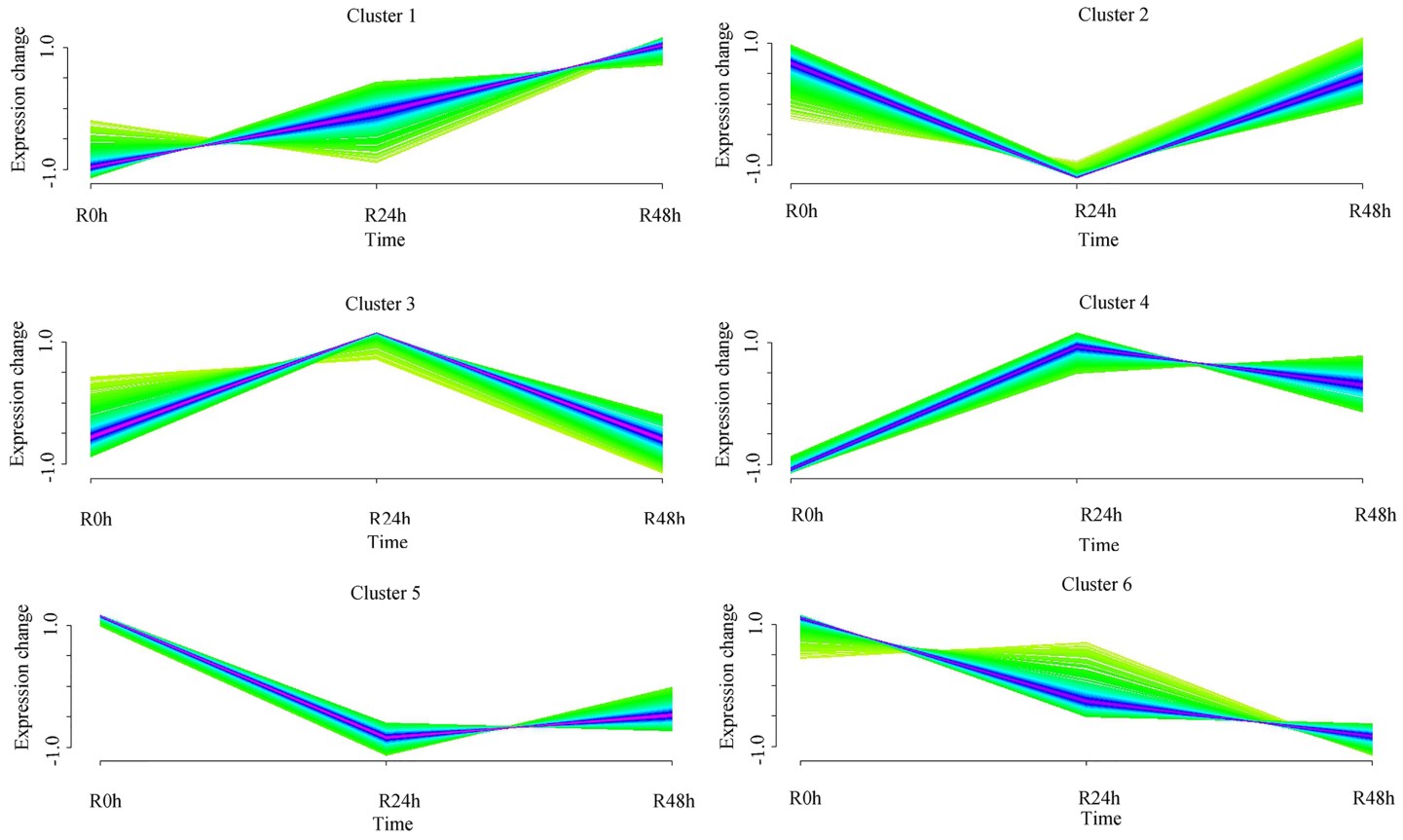

**Figure 4 Clustering and classification of differentially expressed genes of eggplant in response to bacterial wilt over time (0, 24, 48 hpi).** Six trends were determined.

## Widely targeted metabolome analysis

On the basis of UPLC-MS/MS and the metware metabolite database, 661 metabolites were detected (Table S6). The PCA results showed that the first two components could explain 42.23% of the dataset variation (Fig. 5A). Cross-validation indicated the first two components relevant for the classification of variation, which illustrated different directions of response to *R. solanacearum*. The heatmap showed different expression profiles after inoculation with *R. solanacearum* (Fig. 5B). A total of 44 (11 upregulated and 33 upregulated), 25 (six upregulated and 19 upregulated), and 24 (14 upregulated and 10 upregulated) differential metabolites were identified in R-0h_vs_R-24h, R-0h_vs_R-48h, and R-24h_vs_R-48h, respectively (Fig. 5C). Differential metabolites are listed in Tables S7–S9.

A total of 15 and 41 metabolites were upregulated and downregulated, respectively, in at least one time point. A total of two and 11 metabolites were upregulated and downregulated, respectively, at R-24h and R-48h (Fig. 6).

## KEGG of differential metabolites

KEGG classification results showed that the top pathways were metabolic pathways and biosynthesis of secondary metabolites (Fig. S2). Several metabolites were involved in

**Table 2 The kept up-regulated gene at 24 and 48 hpi of eggplant inoculation with _R. solanacearum_.**

| | ID | NR |
|---|---|---|
| Resistance gene | SMEL_001g142870.1 | PREDICTED: TMV resistance protein N-like [_Solanum tuberosum_] |
| | SMEL_005g228040.1 | PREDICTED: pleiotropic drug resistance protein 1-like [_Solanum pennellii_] |
| | SMEL_005g238790.1 | PREDICTED: TMV resistance protein N-like [_Solanum tuberosum_] |
| | SMEL_009g334440.1 | PREDICTED: pleiotropic drug resistance protein 1-like isoform X1 [_Solanum tuberosum_] |
| | SMEL_010g350160.1 | PREDICTED: putative late blight resistance protein homolog R1B-16 [_Solanum lycopersicum_] |
| | SMEL_011g363570.1 | PREDICTED: TMV resistance protein N-like isoform X2 [_Solanum tuberosum_] |
| | SMEL_011g378260.1 | Late blight resistance protein Rpi-amr3 [_Solanum americanum_] |
| | SMEL_011g378630.1 | late blight resistance protein Rpi-amr3 [_Solanum americanum_] |
| WRKY transcriptional factor | SMEL_003g177080.1 | probable WRKY transcription factor 75-like [_Solanum tuberosum_] |
| | SMEL_005g227530.1 | putative WRKY transcription factor 23 [_Capsicum baccatum_] |
| | SMEL_007g291370.1 | PREDICTED: probable WRKY transcription factor 65 isoform X1 [_Solanum tuberosum_] |
| | SMEL_008g310920.1 | PREDICTED: WRKY transcription factor 18-like [_Capsicum annuum_] |
| | SMEL_010g350820.1 | PREDICTED: probable WRKY transcription factor 57 [_Solanum lycopersicum_] |
| | SMEL_010g350830.1 | PREDICTED: probable WRKY transcription factor 57 [_Solanum lycopersicum_] |
| | SMEL_010g354280.1 | PREDICTED: probable WRKY transcription factor 70 [_Solanum tuberosum_] |
| bHLH transcriptional factor | SMEL_001g120030.1 | PREDICTED: transcription factor bHLH18-like [_Solanum tuberosum_] |
| | SMEL_001g128620.1 | PREDICTED: transcription factor bHLH13-like isoform X1 [_Solanum pennellii_] |
| | SMEL_001g134770.1 | PREDICTED: transcription factor bHLH130-like [_Solanum lycopersicum_] |
| | SMEL_004g217680.1 | PREDICTED: transcription factor bHLH93-like [_Capsicum annuum_] |
| | SMEL_005g223990.1 | PREDICTED: transcription factor bHLH13-like [_Solanum tuberosum_] |
| | SMEL_006g251970.1 | PREDICTED: transcription factor bHLH36-like [_Capsicum annuum_] |
| TIFY | SMEL_003g197510.1 | PREDICTED: protein TIFY 10B-like [_Solanum tuberosum_] |
| | SMEL_008g308200.1 | PREDICTED: protein TIFY 5A-like [_Solanum tuberosum_] |
| | SMEL_008g308210.1 | PREDICTED: protein TIFY 5A-like [_Solanum tuberosum_] |
| | SMEL_011g363600.1 | PREDICTED: protein TIFY 4A-like [_Solanum tuberosum_] |

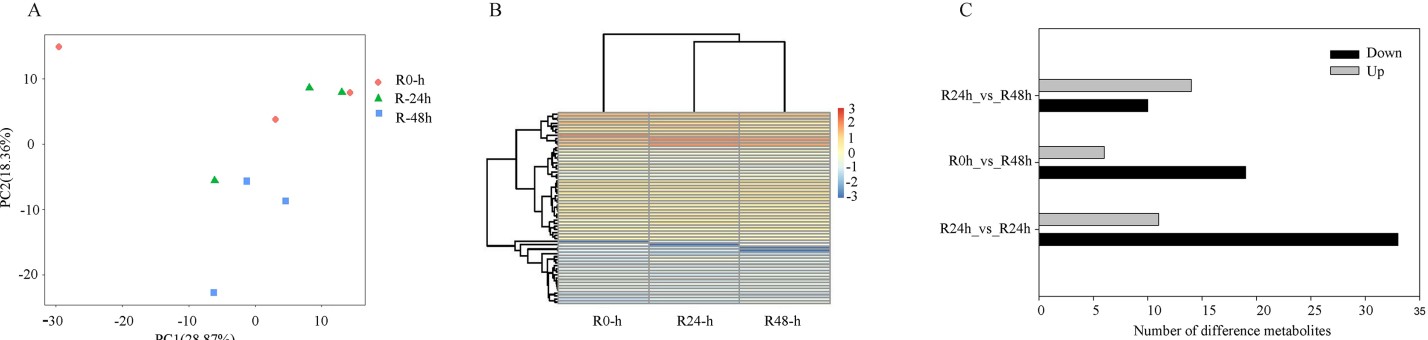

**Figure 5 Differential metabolites of eggplant in response to bacterial wilt.** (A) Principal component analysis of all metabolites detected in root. Data points represent different samples. (B) Clustering analysis and heat map of expression measures of differential metabolites detected in each of the experimental conditions. (C) Numbers of upregulated and downregulated metabolites after inoculation with _R. solanacearum_.

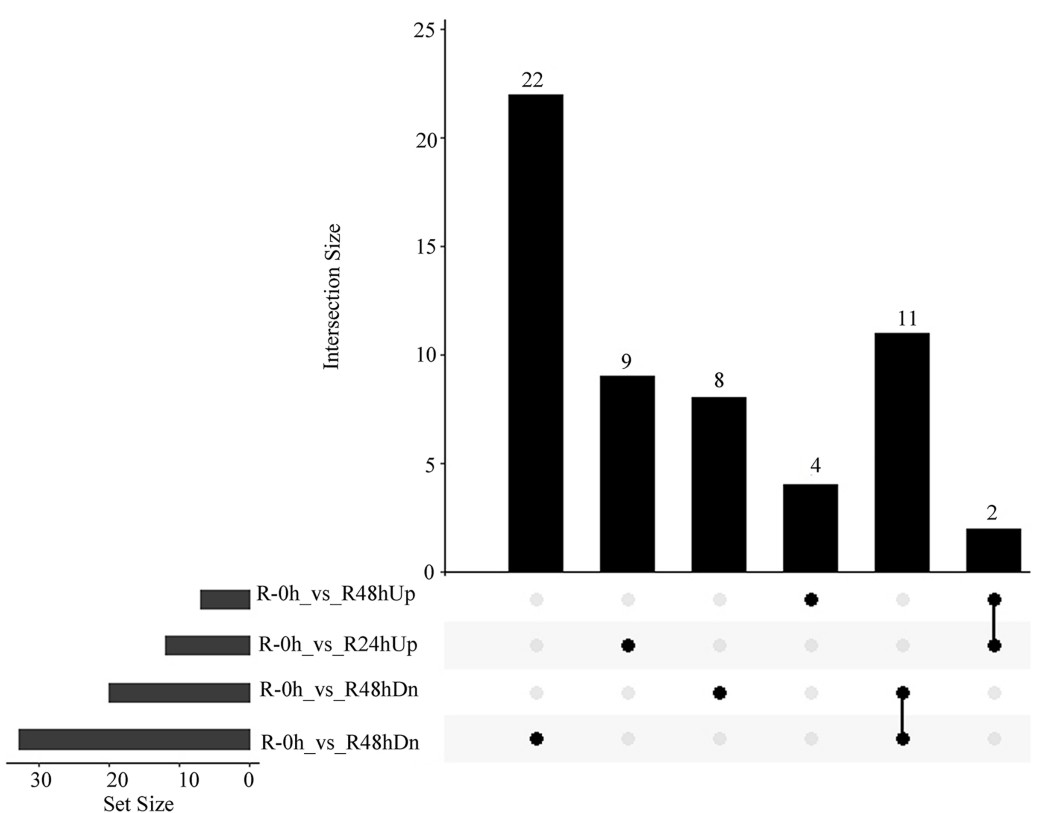

**Figure 6 UpSet plot showing overlap of upregulated and downregulated metabolites of eggplant response to bacterial wilt.**

phenylpropanoid biosynthesis, flavone and flavonol biosynthesis, and plant hormone signal transduction pathway.

## Expression pattern analysis of differential metabolites

A set of metabolites with similar expression patterns was functionally correlated. Six expression patterns were obtained in accordance with differential metabolites data. The expression pattern of the centers of cluster 1 was downregulated at 24 hpi and then upregulated at 48 hpi. However, the expression level at 48 hpi was significantly decreased compared with that at 0 hpi. The expression pattern of the centers of cluster 2 was upregulated at 24 and 48 hpi. The expression pattern of centers of cluster 3 was upregulated at 24 hpi and then significantly downregulated at 48 hpi. The expression level at 48 hpi was significantly decreased compared with that at 0 hpi. The expression pattern of the centers of cluster 4 was downregulated at 24 and 48 hpi. The expression pattern of the centers of cluster 5 was downregulated at 24 hpi and then upregulated at 48 hpi. Ultimately, the expression level at 48 hpi was significantly increased compared than at 0 hpi. The expression pattern of the centers of cluster 6 was downregulated at 24 hpi and at 48 hpi. Ultimately, the expression level at 24 and 48 hpi were not significantly different (Fig. 7 and Table S10). The (−)-Jasmonoyl-L-Isoleucine, the 5′-Glucosyloxyjasmanic acid

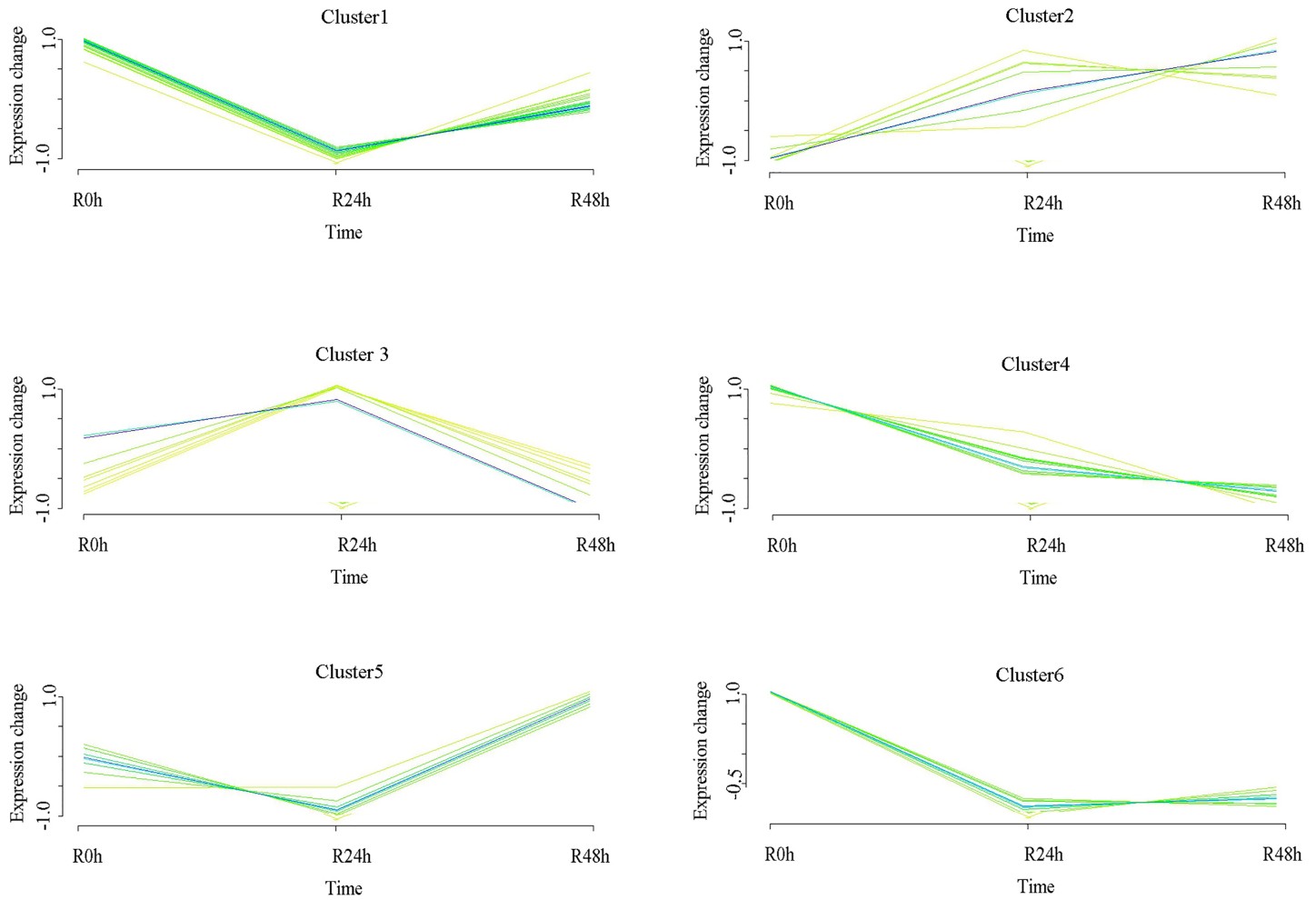

**Figure 7 Clustering and classification of differential metabolites of eggplant in response to bacterial wilt over time (0, 24, 48 hpi).** Six trends were determined.

and 1-O-Salicyl-D-glucose were upregulated at R-24h and R-48h. The salicylic acid was downregulated at R-24h and R-48h.

## Integration analysis of transcriptomic and metabolic datasets

DEGs and differential metabolites were simultaneously assigned to KEGG pathways ($p < 0.05$) to understand the eggplant response to bacterial wilt. The results showed that only alpha-linolenic acid metabolism (ko00592) and plant hormone signal transduction pathway (ko04075) were significantly enriched in the R-0h_vs_R-24h group (Fig. S3). The metabolites involved in these two pathways were JA, ABA, jasmonate, and 9-Hydroxy-12-oxo- 15(Z)-octadecenoic acids. However, 69 genes were involved in these two pathways (Table S9). The ko00592 pathway was the precursor of the JA biosynthesis pathway which ultimately biosynthesizes methyl-jasmonate.

Transcriptome and metabolome data were also compared by the Pearson correlation analysis (Pearson correlation coefficient > 0.8). Gene–metabolite correlation networks were also constructed (Fig. 8). In the R-0h_vs_R-24h group, SMEL_008g298210.1,
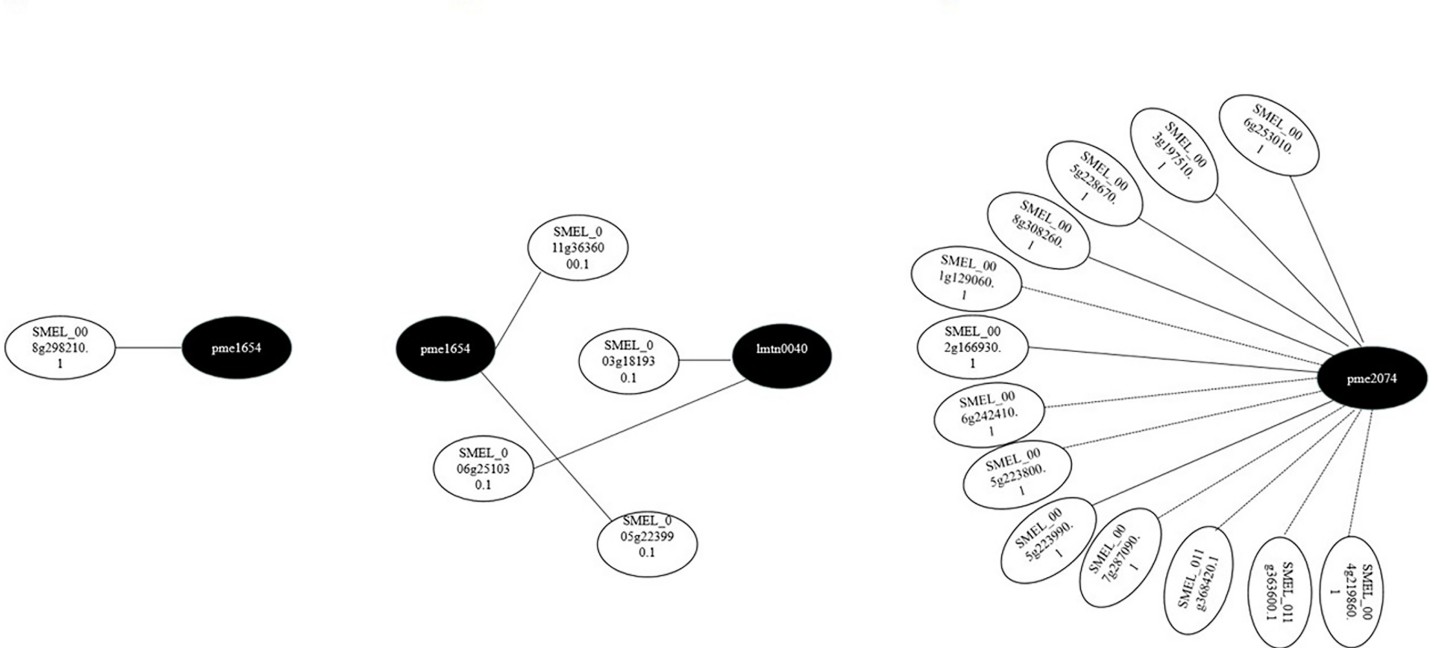

**Figure 8 Gene–metabolite correlation network representing the genes and metabolites involved in the bacterial wilt resistance of eggplant.** (A) Ko00592 network of R-0h *vs* R-24h group. (B) Ko04075 network of R-0h *vs* R-24h group. (C) Ko04075 network of R-0h *vs* R-48h group. White and black dots indicate genes and metabolites, respectively.

SMEL_011g363600.1, and SMEL_005g223990.1 were highly correlated with JA. SMEL_003g183930.1 and SMEL_006g251030.1 were highly correlated with ABA. In the R-0h_vs_R-48h group, 13 genes were highly correlated with (−)-Jasmonoyl-L-Isoleucine. These results suggested that JA might regulate gene expression in the bacterial wilt resistance defense of eggplant.

## DISCUSSION

Breeding of resistant crops is the most efficient process for controlling bacterial wilt. In addition to marker-assisted selection, gene modification and novel genetic editing technologies by CRISP/Cas9 are efficient approaches to develop resistant cultivars. Understanding the resistance mechanism and cloning defense-related genes will accelerate the use of these strategies to develop resistant crops. To the best of our knowledge, the present study is the first to integrate transcriptomic and metabolic techniques and analyze the responses of eggplant to bacterial wilt. These results enhance our understanding of the mechanisms underlying eggplant responses to bacterial wilt. A total of 2,896 DEGs and 56 differences in metabolites were identified after inoculation with *R. solanacearum*. But the replicates between the biological replicates are not ideal. This may be due to problems detection technology.

Plant defense reactions involve a complicated development course of physiological and biochemical changes. The plant hormone signal transduction, MAPK signaling pathway, plant–pathogen interaction pathway, and flavonoid biosynthesis pathway are involved in the bacterial wilt disease response (*Ishihara et al., 2012*; *Chen et al., 2018*; *Dai et al., 2019*;

*Jiang et al., 2019*; *Li et al., 2021a*, *2021c*; *Wei et al., 2021*; *Zhang & Klessig, 2001*). After *R. solanacearum* attack, flavonoids are of prime importance in tomato and *Casuarina equisetifolia* defense responses (*Zeiss et al., 2018*, *2019*; *Wei et al., 2021*). *Peng et al. (2021)* showed that the MAPK signaling pathway, plant-pathogen interaction pathway and glutathione metabolism positively affect the eggplant bacterial wilt resistance. In the present study, the KEGG enrichment analysis of DEGs and differential metabolites showed that, the MAPK signaling pathway, glutathione metabolism, plant-pathogen interaction pathway and flavonoids play an important role in the eggplant response to *R. solanacearum*. Transcriptional factors were such as *WRKY* and *ERF* played key roles in plant defense resistance. *Peng et al. (2021)* demonstrated that *WRKY40*, *WRKY51*, *WRKY53*, *WRKY70*, *BHLH14*, *BHLH111*, *BHLH119*, and *NAC022* play key roles in eggplant bacterial wilt resistance. In the present study, *WRKY40*, *WRKY48*, *WRKY57*, *WRK69*, *WRKY70*, *WRKY75* were upregulated at 24 and 48 hpi. Additionally, in pepper, 4 *WRKY* transcriptional factors were involved in the defense. Two pleiotropic drug resistance protein have been identified in pepper resistant to *R. solanacearm* (*Du et al., 2022*). In the present study, pleiotropic drug resistance protein 1-like and pleiotropic drug resistance protein 1-like isoform X1 were unregulated at 24 and 48 hpi.

Phytohormones, such as SA and JA, play a key role in plant response to biotic stress (*Dong, 1998*; *Feys & Parker, 2000*). Several studies have shown that SA plays an important role in defense against bacterial wilt (*Chen et al., 2016*; *Chen et al., 2018*; *Zeiss et al., 2018*). In the present study, the SA content and genes involved in the SA signaling pathway, such as *NPR1* and *PR1*, differed after the eggplant inoculation with *R. solanacearum*. However, our results indicated that JA increased in the eggplant response to *R. solanacearum*. The contents of JA and JA synthesis precursor, such as (−)-Jasmonoyl-L-Isoleucine, increased after inoculation with *R. solanacearum*. The JA signaling-related gene, such as *JAZ*, *TIFY* and *MYC2*, were upregulated. *Chen et al. (2018)* also showed that the expression levels of *JAZ* and *MYC2* were upregulated after inoculation with *R. solanacearum*. *JAZ* and *MYC2* were master regulators in the JA signaling pathway. *JAZ* and *MYC2* regulate JA-mediated plant immunity. The overexpression of *OsMYC2* increases the early JA-responsive gene expression and the bacterial blight resistance in rice (*Yuya et al., 2016*). Interestingly, riboflavin downregulated at 24 and 48 hpi which suggest that riboflavin may negatively regulate the eggplant bacterial wilt. However, *Peng et al. (2021)* showed that the riboflavin metabolism pathway was enriched in the root and the expression of the riboflavin biosynthesis gene of R genotype was higher than that of S genotype. This may be because the riboflavin regulates the bacterial wilt resistance at late stages such as 3–5 day after inoculation with *R. solanacearum*.

Additionally, integrated transcriptomic and metabolic analyses showed that JA increased in response to bacterial wilt. Based on the literature and our results, we speculated the JA biosynthesis and signaling cascade in eggplant response to *R. solanacearum* (Fig. 9). Although several studies have shown that *Pseudomonas syringae* suppresses host defense responses by activating JA signaling in a COI1-dependent manner (*Katsir et al., 2008*; *Zeng & He, 2010*; *Zhang et al., 2015*; *Zhou et al., 2015*; *Yang et al., 2019*), these results showed that JA negatively regulates the *Pseudomonas syringae* resistance. However, the result of present

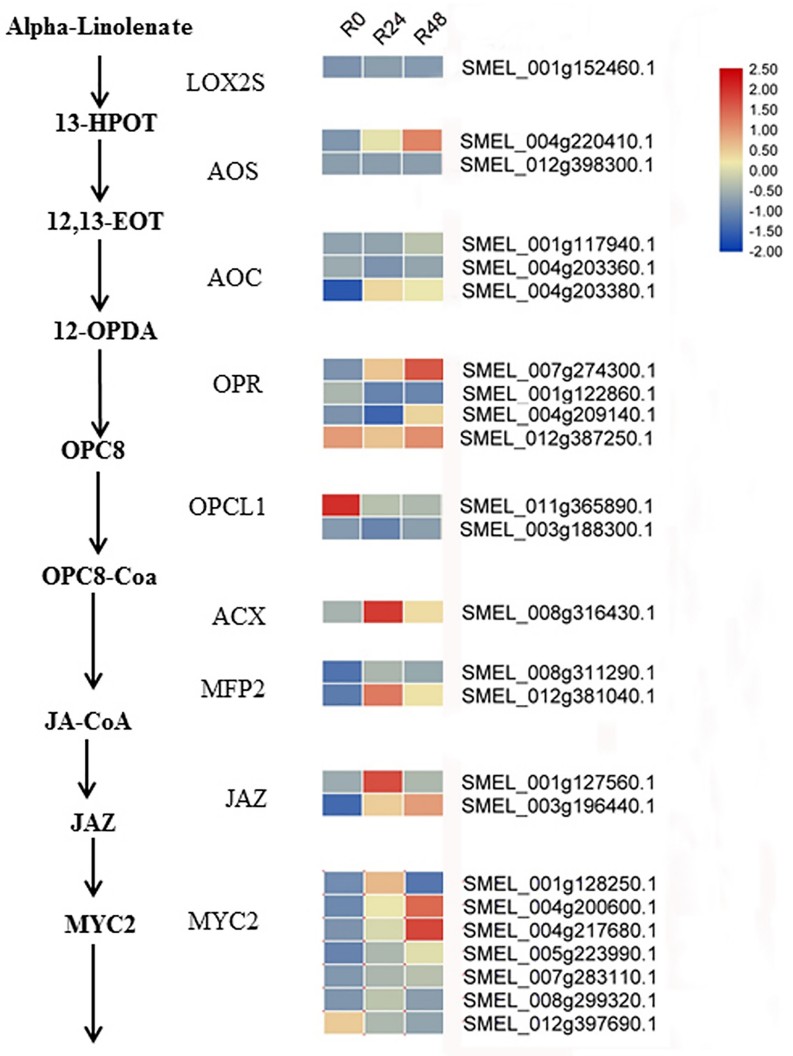

**Figure 9 Heat maps of genes involved in the JA biosynthesis and signaling cascade after inoculation with _R. solanacearum._** The scale represents the z-score normalized FPKM values.

study suggests that JA may positively regulate the bacterial wilt resistance. In tomato, the JA-dependent signaling pathway is required for biocontrol agent-induced resistance against _R. solanacearum. Jiang et al. (2019)_ showed that silicon treatment increases the content of SA, JA, SA and JA-related genes to improve the bacterial wilt resistance of tomatos. In future studies, the functions of JA in the response of eggplant to _R. solanacearum_ will be analyzed.

## CONCLUSIONS

Integrated transcriptomic and metabolomic analyses generated a set of data to reveal the eggplant defense response to bacterial wilt. Defense responses include biosynthesis of flavones, flavonoids and phytohormones. The gene expression and metabolic networks

identified in this study provide new insights into the mechanisms of the induced defense response in eggplant. Our results will remarkably improve our knowledge of the bacterial wilt resistance mechanism of eggplant and provide clues for the development of resistant eggplant varieties.

### Funding

This research was supported by the National Natural Science Foundation of China (No. 31872117), the Central Public-interest Scientific Institution Basal Research Fund (No. 1630062022003), the Central Public-interest Scientific Institution Basal Research Program for Scientific Research Innovation Team (No. 1630062017014), and the National Natural Science Foundation of Hainan Province (No. 321RC632). The funders had no role in study design, data collection and analysis, decision to publish, or preparation of the manuscript.

### Grant Disclosures

The following grant information was disclosed by the authors:
National Natural Science Foundation of China: 31872117.
Central Public-interest Scientific Institution Basal Research Fund: 1630062022003.
Central Public-interest Scientific Institution Basal Research Program for Scientific Research Innovation Team: 1630062017014.
National Natural Science Foundation of Hainan Province: 321RC632.

### Competing Interests

The authors declare that they have no competing interests.

### Author Contributions

- Xi Ou Xiao conceived and designed the experiments, prepared figures and/or tables, and approved the final draft.
- Wenqiu Lin performed the experiments, analyzed the data, authored or reviewed drafts of the article, and approved the final draft.
- Enyou Feng analyzed the data, authored or reviewed drafts of the article, and approved the final draft.
- Xiongchang Ou performed the experiments, authored or reviewed drafts of the article, and approved the final draft.

### Data Availability

The data is available at GenBank: PRJNA837016.

### Supplemental Information

Supplemental information for this article can be found online at http://dx.doi.org/10.7717/peerj.14658#supplemental-information.

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
