# Peer review of "Transcriptome and metabolome response of eggplant against Ralstonia solanacearum infection"

_PeerJ, doi:10.7717/peerj.14658_

## Round 0.1 · original submission · Major Revisions

Your manuscript has been now assessed by three reviewers. I find your article very interesting and potentially relevant for the community. However valid concerns have been raised. I would therefore consider publication of your article provided it undergoes a revision tackling the issues raised.

Reviewer 1 ·

Basic reporting

The paper by Xiao et al presents a transcriptome and metabolome level study of eggplant response to the bacteria Ralstonia. The article is novel and interesting results emerge. However, the discussion of these results is very limited. The authors should make an effort to broaden the discussion, expand the study of the panels of DEG genes and metabolites involved in the trait of interest (since they only focused on one group, phytohormones), and discuss the results with respect to those previously published by other authors.
I attach comments:
- In M&M authors specify that the genotype used is NY-1 (resistant). However, in the results section authors talk about two genotypes, one susceptible and one resistant. If a susceptible was also used, indicate it in M&M and provide information about it.
- Include the ID and version of the reference genome used, to ensure reproducibility.
- Line 114 says: "After trimming adapter sequences and removing low-quality reads by using the FastQC tool". Correct, since FastQC does not perform cleaning (it only allows a visual inspection of read quality). Indicate the software actually used and the parameters applied for read cleaning.
- Line 179 only says: "The sample size is 7.40". Please complete and discuss.
- Figure 2b: the heatmap does not contain all the DEG genes, does it? Is it just a selection? The low quality of the figure does not allow to see well, but a priori it would seem that the tree has few branches, which represents a few genes. Please improve the quality of the figure. And if not all the DEG genes are represented, but only a few, please indicate: on what basis were they selected?
- The authors should expand the result and discussion sections aimed at interpreting the output of the soft clustering analysis (Figure 4). Also, add references to the figure, to make it self-explanatory.
- In Figure 5A (PCA), biological replicates are not seen to cluster (as seen in Figure 2A). The authors should discuss what this might be due to.
- In line 43, the authors say: "Many QTLs (quantitative trait locus) resistant to bacterial wilt are identified in different plants, such as eggplant...". That paragraph gives the idea that such information is to be included/analyzed in the article, in relation to the data obtained. However, this is not observed. The authors could include a co-localization study of the QTL reported in the literature as associated with resistance to this disease, with the DEG genes/metabolites identified here.
- The authors should broaden the discussion. They focused only on a small group of genes (Phytohormones), when DEGs are many more. For example, one thing they should explore, as they are working on disease resistance, is whether or not there are NLR DEGs. They should also include comparison with DE genes identified in other previously published studies of eggplant response to this pathogen (e.g., Chen et al 2018).
- Authors should include the accession ID to the data in the manuscript (PRJNA837016).
- In figure S2, sort the references alphabetically (or in some logical order).
- Improve the quality of Figure 9. Also, I don't quite understand where the heatmap data for time 0 comes from? What were they compared against, to know if they were UP or Down? Please expand.

Experimental design

All my comments were previously detailed.

Validity of the findings

All my comments were previously detailed.

Additional comments

All my comments were previously detailed.

Reviewer 2 ·

Basic reporting

In this manuscript titled “Transcriptome and metabolome response of eggplant
against Ralstonia solanacearum infection” by Xiao et al., the authors reported the DEGs and differential metabolites in eggplant after Ralstonia infection. To improve the quality of this manuscript, and to ensure the results are reliable and reproducible, there are some concerns to be addressed.
1. The manuscript can be organized in a better way. There are some sentences seem to randomly appear in some parts of the manuscript. For example, line 103-105, line 179, line 413. The writing needs to be significantly improved. Species name needs to be italicized.
2. In Fig 1, it would be helpful if the authors can include phenotypic images of the R and S eggplant at the 0, 24, 48 hpi, which are the timepoints of collecting samples for RNA-seq.
3. The authors described Fig 3 and Fig 6 as Venn diagrams. However, Fig 3 and Fig 6 are not Venn diagrams.
4. The authors need to explain the definition of each cluster in the Fig 4 and Fig 7, describe the method, and interpret the results in more details.
5. In the introduction, the authors need to explain the difference between the R and S genotype of eggplants, and introduce the current study about the mechanism of Ralstonia resistance in the R genotype.
6. The authors need to compare their findings with previous transcriptome and metabolome studies upon Ralstonia infection in more details in the discussion. The authors included the Zhao et al., 2019 and Zuluaga et al., 2015 paper in the reference list, but these two papers were not mentioned in the main text. The authors need to introduce or discuss these two papers.

Experimental design

1. First, in the DEG analyses, instead of comparing three time points with each other, the S genotype obviously will be a better control for each time point. But the authors did not use the S genotype as a control for RNA-seq and metabolic analyses.
2. Additionally, the authors need to describe the methods more accurately and in more details. Specifically, these following concerns need to be addressed.
(1) The authors need to describe the S genotype in the “Plant materials” section.
(2) Please include detailed information of the eggplant genome assembly used, including assembly accession number, assembly level and species name.
(3) The authors need to specify the experiment conditions in the header of each sample/run in the deposited RNA-seq data in the NCBI SRA database.
(4) Line 101, what is “LuYour3415”?
(5) Line 115-117, the DESeq2 package in R cannot directly take the output from HISAT2 as the input. Please describe the steps accurately and in more details. And specify the version of R.
(6) Line 118, FDR and FDR adjusted P value are different concepts. Please be specific.
(7) The authors need describe the definition and measurement of the disease index (DI), which was shown in Fig 1A.
(8) The authors need to describe the methods used for constructing the gene–metabolite correlation networks shown in Fig 8.
(9) In Fig 2B and Fig. 9, the data points in the heatmap seems to be ratio of expression calculated from DESeq2. In each column, what was the control group, what was the experimental group? Especially for 0 h, if this is the ratio of expression, what was it compared to? Similar issue for Fig 5B.

Validity of the findings

There are some critical information missing in the materials and methods section and figure legend, especially the DEG analyses, so the validity cannot be fully accessed in the current form. I will be willing to assess the validity of the findings in the next submission, if the necessary information is provided.

Additional comments

Some of the references were referred by the authors first name, instead of last name. For example, line 442-443, “Xi-ou X, Wenqiu L, Zuo C, Chunxiang Z, Hui J, and Huafen Z. 2021. Wide-host Vector pBBR1MCS2-Tac-EGFP Suitable for the Labeling of Ralstonia solanacearum. Chinese Journal of Tropical Crops 42:1700-1705.” And line 420-422.
Some of the references included DOI, some did not. Please be consistent.
Please proof-read the manuscript carefully to fix any additional formatting issues, including the inconsistency in the reference section.

Reviewer 3 ·

Basic reporting

For review please see attached pdf

Experimental design

For review please see attached pdf

Validity of the findings

For review please see attached pdf

Annotated reviews are not available for download in order to protect the identity of reviewers who chose to remain anonymous.

---

## Round 0.2 · Major Revisions

According to the reviewer's comments a large proportion of concerns remained to be addressed to improve the quality of this manuscript.

Reviewer 2 ·

Basic reporting

In the revised version of “Transcriptome and metabolome response of eggplant against Ralstonia solanacearum infection”, the authors have addressed part of the reviewers’ concerns. However, a large proportion of concerns remained to be addressed to improve the quality of this manuscript.
1. As the reviewer pointed out previously, Figure 3 and Figure 6 are not Venn diagrams. The authors replied in the rebuttal letter claiming these to be “other types of Venn diagrams”. These are UpSet plots, which are different from Venn diagrams. I recommend the authors to read the information from the following website to get more familiar with the differences between UpSet plots and Venn diagrams.
http://gehlenborglab.org/research/projects/upsetr/
https://upset.app/#upset-vs-venn-diagrams
2. The reviewer suggested the authors to include phenotypic images of the R and S eggplant at the 0, 24, 48 hpi, which are the timepoints of collecting samples for RNA-seq (See reviewer 2, basic reporting, point 2). The authors replied, “The eggplant showed the wilt phenotypic generally at the 5 days after inoculation with R. solacearum”, which does not seem to be a resolution for this suggestion. Even though the eggplants may not show wilting symptoms at 24 and 48 hpi, these qualitative phenotypic images are important, because these datapoints were used to collect the sample for the RNA-seq in this study.
3. The reviewer suggested the authors to explain the definition of each cluster in the Figure 4 and Figure 7 (See reviewer 2, basic reporting, point 4). The authors replied that “The Fig 4 and Fig 7 were analyzed by the Mfuzz R package and the cluster is definited by the Mfuzz R package. We don’t need to definite each cluster.” Ture, the Mfuzz R package defined each cluster, but I suggest the authors to read the help document of the Mfuzz R package to learn how were these clustered, and explain that in the result section, not just displaying the output from an R package without enough explanation. Without proper explanation and interpretation, these figures do not contribute to the findings of this study.
4. As the reviewers pointed out previously, the manuscript needs to be better organized and the writing needs to be significantly improved. Unfortunately, there are many typographical and formatting issues. And some spaces were missing, or placed in the wrong place. For example:
Line 113-115, “The statistical power … sequencing depth were 7.55”, should be moved to the end of the RNA-seq section.
Line 124, “fastp(v 0.19.3)”.
Line 128, “featureCounts( v1.6.2)”.
Line 161, “a a specific set of multiple reaction monitoring transitions was monitored”.
Please perform rigorous proofreading to correct any remaining issues, including punctuations, not just the ones listed by the reviewers.
5. The reviewer requested the authors to include detailed information of the eggplant genome assembly used, including assembly accession number, assembly level and species name (see reviewer 2, basic reporting, point 8.2; and reviewer 1, point 2). The authors stated “The clead reads were aligned to the eggplant reference genome (Barchi et al., 2019) using HISAT2(version 2.1.0).” Please specifically list the assembly version and accession number in the methods section, not just list a reference. Also, “clead reads” should be “cleaned reads”.
6. Reviewer 1 suggested the authors to discuss the reason causing the biological replicates not clustering in the PCA (Figure 2A and 5A). The authors replied in the rebuttal letter, not the manuscript. The authors should add this to the discussion section of the manuscript, not just the rebuttal letter.
7. In Figure 9, the scale represents the z-score normalized FPKM values, not the logarithm of FPKM values. Again, I recommend the authors to read the help document of the heatmap package they used. And the authors should describe which heatmap package was used, in the methods section.

Experimental design

See above.

Validity of the findings

See above.

---

## Round 0.3 · Minor Revisions

Please address the following items from the Section Editor:

"Abstract and discussion overstates the conclusions. This is a correlational study that found an increase in JA signaling genes and metabolites after innoculation with Ralstonia. This does not prove that JA regulates or plays a "key role" in resistance; to do so would require manipulation of JA signaling or synthesis. I was surprised that although sensitive and resistant varieties are described, the transcriptome and metabolite studies are only done on the resistant variety, so all we have is a time course and we don't even know if the RNA and metabolites identified as changing in response to infection are less induced in the sensitive strain. Thus, the wording in the abstract and discussion must be changed to something like "JA signaling increased in response to" or "these results suggest that".

+++ Gene names need to be italicized throughout

+++ Figure text is too small to read in many places

+++ Additionally there are many grammatical errors. A few examples line 301 "Transcription factors were such as..."

+++ 302 "Peng...showed that that WRKY...were identified which played a key role"

+++ line 216 "was not significant difference"

+++ line 220 " the expression level at 48 hpi was significantly increased than that at 0hpi." and many others. I recommend a professional editing service."

---

## Round 0.4 · accepted · Accept

Authors incorporated all of the suggestions needed.